



# The Estimation of Path Integrated Attenuation for the EarthCARE Cloud Profiling Radar

Susmitha Sasikumar[1], Alessandro Battaglia[1,2], Bernat Puigdomènech Treserras[3], and Pavlos Kollias[3,4]

[1]Department of Environment, Land and Infrastructure Engineering , Polytechnic University of Turin, Turin, Italy
[2]Department of Physics and Astronomy, University of Leicester, Leicester, UK
[3]Department of Atmospheric and Oceanic Sciences, McGill University, Montreal, QC Canada
[4]School of Marine and Atmospheric Sciences, Stony Brook University, Stony Brook, NY, USA

**Correspondence:** Susmitha Sasikumar (susmitha.sasikumar@polito.it)

**Abstract.** The joint ESA and JAXA Earth Cloud, Aerosol and Radiation Explorer (EarthCARE) satellite, launched on 28 May 2024, carries the first spaceborne 94 GHz Cloud Profiling Radar (CPR) with Doppler velocity measurement capability. As a successor to the highly successful NASA CloudSat CPR, the EarthCARE CPR offers an additional 7 dB of sensitivity largely due to its larger antenna size (2.5 m vs. 1.8 m) and lower orbit (400 vs. 700 km), and a receiver point target response that

significantly improves our ability to detect clouds in the lowest km of the atmosphere. The EarthCARE CPR measurements can also be indirectly used to estimate the Path-Integrated Attenuation (PIA, in dB), a measure of two-way attenuation caused by hydrometeors by quantifying the depression in the measured normalized radar cross section (NRCS) relative to a reference NRCS in the absence of hydrometeors. PIA is a key constraint for improving the accuracy of cloud and precipitation retrievals.

This paper presents the PIA estimation methodology currently operationally implemented in the EarthCARE CPR L2A C-

PRO data product. The retrieval approach follows a hybrid strategy, where the reference unattenuated NRCS is either estimated using calibration points surrounding the cloudy profile where PIA is estimated or a model-based estimation that uses a geophysical model that calculates NRCS as a function of wind speed and sea surface temperature (SST). The methodology provides a full characterization of the uncertainty in PIA estimates and is expected to lead to improved estimates of PIA compared to the methodology adopted for the CloudSat CPR. This method is particularly useful in PIA estimation in the commissioning phase

of the mission, as it is robust for radar miscalibration and bias of gas attenuation or NRCS modeling.

## 1 Introduction

When operated from a spaceborne platform in a nadir-looking configuration, radar systems transmit pulses towards the Earth's atmosphere and receive backscattered signals from atmospheric targets. As the radar pulse traverses the atmospheric column, it experiences two-way attenuation due to two primary mechanisms: (1) absorption by atmospheric gases, and (2) scattering

and absorption by hydrometeors such as cloud and precipitation particles. This cumulative signal loss is referred to as Path-Integrated Attenuation (PIA). The total PIA can be decomposed into two components: gaseous attenuation ($PIA_{gas}$) and hydrometeor attenuation ($PIA_{hydro}$), as expressed in Eq. (1) (Lebsock et al., 2011)

$$PIA = PIA_{gas} + PIA_{hydro}. \tag{1}$$





The two terms are particularly large for millimeter wavelength radars that have traditionally been used to study clouds and precipitation systems (Kollias et al., 2007). The two-way gaseous attenuation component ($PIA_{gas}$) is estimated using atmospheric absorption models based on thermodynamic profiles (see Sect. 2.1), whereas the $PIA_{hydro}$ accounts for two-way integrated extinction caused by hydrometeors, assuming negligible effects from multiple scattering (Battaglia et al., 2010, 2011), and is expressed as:

$$\text{PIA [dB]} = \frac{20}{\log(10)} \int_0^H k_{\text{ext}}(z)\,dz \qquad (2)$$

where $k_{\text{ext}}$ is the height-dependent extinction coefficient (in m$^{-1}$) due to cloud and precipitation particles (Haynes et al., 2009).

At 94 GHz, the Earth's surface acts as a strong radar reflector, returning signals with an intensity that is often several orders of magnitude greater than that from atmospheric targets. When attenuating hydrometeors (e.g., rain, snow, or cloud particles) are present in the radar beam, they reduce the strength of the surface return. This diminished surface echo, or surface signal depression, can be analyzed to quantify the attenuation introduced by hydrometeors (Meneghini and Kozu, 1990; Meneghini et al., 2004).

At 94 GHz, attenuation by cloud liquid water is well described by Rayleigh scattering theory, resulting in an approximately linear relationship with the liquid water path (LWP) and exhibiting moderate sensitivity to temperature. In contrast, attenuation caused by precipitation particles must be modeled using Mie scattering theory, as the interaction of radar waves with larger hydrometeors depends on both temperature and the drop size distribution (DSD) of the precipitation. Despite these complexities, the total Path-Integrated Attenuation (PIA) is largely dominated by the column-integrated liquid water content, making it a robust proxy for estimating LWP (Lebsock et al., 2011; Battaglia et al., 2020; Lebsock et al., 2022).

Numerous studies have highlighted the effectiveness of using PIA for improving rainfall estimation. Traditional ground-based radar rainfall retrieval algorithms often rely on the Rayleigh approximation, which assumes that raindrops are small relative to the radar wavelength. These methods typically employ an assumed DSD to derive a simplified relationship between radar reflectivity and rainfall rate. However, for space-borne radars operating in the microwave spectrum, relying solely on reflectivity is insufficient due to the significant influence of attenuation. High-frequency radars, such as the CloudSat and EarthCARE 94 GHz radars experience significantly greater attenuation than lower-frequency radars for the same rain intensity (Haynes et al., 2009). Hence, for high frequency radars, PIA is a useful measurement in rainfall and LWP retrieval (Tridon et al., 2020). To address this, (L'Ecuyer and Stephens, 2002) proposes a retrieval method specifically designed for attenuating radars, emphasizing the use of PIA or estimates of LWP as constraints. A comparative analysis between 14 GHz (Ku-band) and 94 GHz (W-band) radars demonstrates that reflectivity-based rain rate estimates at 94 GHz become highly uncertain beyond 1 mm h$^{-1}$, whereas the 14 GHz radar provides reliable estimates up to 40 mm h$^{-1}$. When LWP is incorporated as a constraint, the retrieval accuracy improves significantly, underscoring the critical role of PIA in rainfall retrieval for high-frequency radar systems. The CloudSat warm rain retrieval algorithm utilizes this method and employs a hybrid approach, using reflectivity-based retrieval at lower rain rates and switching to an attenuation-based method at higher rain rates, where





attenuation becomes more pronounced. Quantitative analysis of the algorithm reveals that this transition between reflectivity-dominant and attenuation-dominant retrieval occurs within the rain rate range of approximately 0.1 to 0.5 mm h$^{-1}$ (Lebsock and L'Ecuyer, 2011).

Given the importance of PIA in microwave remote sensing, its accurate estimation largely depends on the reliable characterization of the effective surface backscattering cross section ($\sigma_{0e}$). Over the ocean, $\sigma_{0e}$ can be parameterized as a function of incidence angle, sea-surface temperature, and wind speed (Li et al., 2005). In contrast, over land, $\sigma_{0e}$ is highly variable and depends on factors such as vegetation type, soil moisture, and terrain roughness, making characterization of $\sigma_{0e}$ far more difficult (Haynes et al., 2009). Consequently, PIA estimation and PIA-based rainfall retrievals are generally restricted to oceanic
regions, where geophysical models can provide robust estimates of $\sigma_{0e}$. Section 2.4, explores in detail how $\sigma_{0e}$ over the ocean varies with wind speed and assess the ability of different geophysical models (Li et al., 2005) to reproduce the EarthCARE $\sigma_{0e}$ climatology.

CloudSat's 2C-PRECIP-COLUMN product (2C-PRECIP-COLUMN Product Description, 2018) employs two complementary methods for estimating PIA. In regions with scattered clouds, where adjacent clear-sky observations are available, the
Surface Reference Technique (SRT) is used. This method estimates PIA by interpolating the clear-sky surface backscattering cross section from nearby cloud-free profiles over the cloudy areas. However, in the presence of extensive, continuous cloud cover, where no nearby clear-sky profiles are available, SRT cannot be applied. In such cases, PIA estimation relies entirely on geophysical models (see Sect. 4).

In this paper, we propose a PIA retrieval scheme that mirrors CloudSat's dual-path strategy but is tailored to the early
operational phase of EarthCARE, when instrument calibration is still evolving. Section 2 outlines the methodology in detail. The modeling of gaseous attenuation used to derive $PIA_{gas}$ is detailed in Sect. 2.1, while Sect. 2.2 describes the procedure for estimating the normalized radar cross section (NRCS). Section 2.3 explains criteria used in selection calibration points for the SRT. Section 2.4 explores how the $\sigma_{0e}$ varies with wind speed and assess the performance of different geophysical models. Section 3 illustrates our retrieval's performance through several case studies. Comparative analysis with CloudSat's
PIA estimates is provided in Sect. 4 and Sect. 4.1, highlighting the consistency and potential advantages of the proposed method.

## 2 Methodology

Hereafter the $PIA_{hydro}$ ($PIA_{gas}$) indicates the two-way PIA associated to the hydrometeors (atmospheric gases). The unknown is $PIA_{hydro}(x)$ at a given location $x$ (where clouds and/or precipitation are presents). The $\sigma_{0e}$, which is effective
back scattering cross section represents the expected surface backscatter signal under no atmospheric attenuation (gases and hydrometeors). The measured NRCS at point $x$ denoted by $\sigma_{0m}(x)$ will be related to the $\sigma_{0e}$ at the same point by:

$$\sigma_{0m}(x) \quad = \quad \sigma_{0e}(x) - PIA_{gas}(x) - PIA_{hydro}(x). \tag{3}$$





If in the vicinity of $x$ there is a location $x_1$ characterized by calibration condition, then:

$$\sigma_{0m}^{cal}(x_1) \quad = \quad \sigma_{0e}(x_1) - PIA_{gas}(x_1) \tag{4}$$

Section 2.3 provides a detailed explanation on how calibration points are chosen. The effective surface normalized radar cross section can be either derived from a geophysical model with $\sigma_{0e}$ being a function of wind speed and SST (Li et al., 2005) or from measured NRCS by correcting for gas attenuation. The $PIA_{gas}$ term can be derived from gas attenuation models (given the atmospheric thermodynamic profile; see Sect. 2.1). With these components, the $PIA_{hydro}$ can be inferred by inverting Eq. (3) as:

$$PIA_{hydro}(x) = \underbrace{\sigma_{0e}(x) - PIA_{gas}(x)}_{\sigma_0^{gas}(x)} - \sigma_{0m}(x) \tag{5}$$

On the other hand, by subtracting Eq. (4) from Eq. (3) an alternative relationship for computing $PIA_{hydro}(x)$ is obtained as :

$$PIA_{hydro}(x,\ x_1) = \underbrace{[PIA_{gas}(x_1) - PIA_{gas}(x)] + [\sigma_{0e}(x) - \sigma_{0e}(x_1)] + \sigma_{0m}^{cal}(x_1)}_{\sigma_0^{gas}(x,\ x_1)} - \sigma_{0m}(x) \tag{6}$$

where in both Eq. (5) and Eq. (6) $\sigma_0^{gas}(x)$ and $\sigma_0^{gas}(x,\ x_1)$ (respectively) represent two ways to estimate the NRCS that would be measured at $x$ in the absence of hydrometeors (but with the presence of gases).

The advantage of computing PIA by Eq. (6) is that only differences appear on the right hand side of Eq. (6), which makes the estimate very robust for radar miscalibration [affecting the values of $\sigma_{0m}$ but not of the difference $(\sigma_{0m}^{cal}(x_1) - \sigma_{0m}(x)]$ and for biases of the gas attenuation or in the $\sigma_{0e}$ estimation. For any given cloudy/rainy profile at position $x$, there may be multiple neighboring calibration points. If such $N$ points are over the contiguous ocean free of ice they can be used as calibration points. Eq. (6) can be generalized to:

$$PIA_{hydro}(x,\ x_1,\ x_2 \ldots x_N) = \cfrac{\sum_{i=1}^{N} w_i \left\{ \overbrace{[PIA_{gas}(x_i) - PIA_{gas}(x)] + [\sigma_{0e}(x) - \sigma_{0e}(x_i)] + \sigma_{0m}^{cal}(x_i)}^{\sigma_0^{gas}(x,\ x_i)} \right\}}{\underbrace{\sum_{i=1}^{N} w_i}_{\sigma_0^{gas}(x,\ x_1,\ldots,\ x_N) \equiv \tilde{\sigma}_0^{gas}(x)}} - \sigma_{0m}(x) \tag{7}$$

In the PIA estimation algorithm used in the EarthCARE product, an optimal number of $N = 5$, calibration points is used to ensure that calibration points remain sufficiently close, as more distant points have negligible influence (see Sect. 2.3 for a detailed explanation of calibration point selection).

    Each $\sigma_0^{gas}(x, x_i)$ term in Eq. (7) is assigned a weight $w_i$ that reflects the uncertainty introduced when using anchor point $x_i$. 110 This uncertainty primarily depends on two factors: the distance to the calibration point ($d(x, x_i)$) and the wind speed at that cloudy profile ($x$). To quantify this uncertainty, a large set of clear-sky profiles with measured NRCS ($\sigma_0^{gas}$) and wind speed is compiled. Clear-sky profiles are chose based on "profile_class" product, from the Level 2 C-PRO FMR dataset (Kollias et al., 2023). For each profile $x$, all other profiles $x_i$ are used as calibration points to estimate $\sigma_0^{gas}(x, x_i)$ following Eq. (7).



The difference between the estimated and measured NRCS at $x$ defines a residual that reflects the error associated with using calibration point $x_i$. The residual is computed as:

$$\Delta\sigma_0(d(x,x_i),u(x)) = \Delta\sigma_0(x,x_i) = \sigma_0^{gas}(x,x_i) - \sigma_0^{gas}(x) \tag{8}$$

These residuals are binned by distance and wind speed and for each bin, the standard deviation of residuals is computed to represent the uncertainty:

$$\sigma_{\text{uncer}}(d_k,u_j) = \text{std}\left\{\Delta\sigma_0(d,u) \,\big|\, d \in b_{d_k},\, u \in b_{u_j}\right\} \qquad for \quad k=1,\ldots N_d; \quad j=1,\ldots N_u \tag{9}$$

where $b_{d_k}$ and $b_{u_j}$ denote the k-th distance bin and the j-th wind speed bins, respectively.

In the PIA estimation with Eq. (7), the weights $w_i$ assigned to each calibration point $x_i$ are defined as the inverse of the squared uncertainty associated with each calibration point ($\sigma_{\text{uncer}}(d(x,x_i),u(x))$, such that points with lower uncertainty contribute more strongly to the estimate: ($w_i = 1/(\sigma_{\text{uncer}}(d(x,x_i),u(x)))^2$). Figure 1 represents the PIA uncertainty look-up table built by the described method and Fig. 2 is a schematic depiction of the described PIA estimation methodology.

As an alternative to the clear-sky interpolation method, PIA can be estimated directly using Eq. (5), where $\sigma_{0e}$ is derived from a geophysical model or from a climatology-based derived lookup table that estimates $\sigma_{0e}$ as a function of wind speed and SST, in case the uncertainty by clear sky interpolation is large. This approach is referred to as the model-driven method or Wind/SST method. In Fig. 1, the black curve delineates the region beyond which the model-driven method becomes more reliable than interpolation. In evaluating the error associated with the model-driven method, $\sigma_{0e}$ is derived from a climatologically constructed look-up table based on clear-sky NRCS measurements, corrected for gaseous attenuation, as a function of sea surface temperature (SST) and wind speed over the period June 2024 to June 2025. For each SST-wind speed bin, the mean and standard deviation of $\sigma_{0e}$ are computed. The uncertainty associated with $\sigma_{0e}$ as a function of wind speed is estimated by calculating the weighted mean of standard deviations across SST bins. For wind speeds between 4-15 m/s, and calibration point distances ranging from $\approx 200$ km to $\approx 100$ km respectively, interpolation generally yields lower uncertainties, often below 1 dB. However, as distance increases, the interpolation error rises and the model-driven approach becomes preferable. As wind speed reduces, the interpolation generally yield lower uncertainty even for larger calibration point distances.

Hence, during periods when the radar is well-calibrated, a hybrid approach can be employed, combining the interpolation method using $N$ calibration points (Eq. (7)) and model driven method (Eq. (5)).

The total uncertainty in the two-way PIA estimate at a given location $x$ arises from two main sources, uncertainty in estimating the $\tilde{\sigma}_0^{gas}(x)$, and from the inherent measurement error in radar reflectivity. The first component is estimated from weight associated with each calibration points as each weight $w_i$ corresponds to the inverse of the variance associated with the calibration point at $x_i$, leading to a total uncertainty expressed as:

$$\sigma_{gas}^{uncer}(x) = \left(\sum_{i=1}^{N} w_i\right)^{-1/2} \tag{10}$$

In addition to this methodological error, inherent measurement error in radar reflectivity which depends on the signal-to-noise ratio (SNR) and the number of independent samples ($n_{\text{samples}}$) also contributes to the overall uncertainty in the PIA estimate.




This error is analytically estimated as (Doviak and Zrnić, 1993):

$$\sigma_z[dB] = 10 \log_{10}\left(1 + \frac{1 + \frac{1}{SNR}}{\sqrt{n_{\text{samples}}}}\right) \tag{11}$$

The number of samples is estimated from the pulse repetition frequency (PRF), the integration length ($L_{int}$), and the ground satellite velocity ($V_{sat}$) as:

$$n_{\text{samples}} = \text{PRF}\,\frac{L_{int}}{v_{sat}} \tag{12}$$

For the EarthCARE Level 2 C-PRO FMR dataset, the integration length is 1 km, the PRF varies between 6100 Hz and 7500 Hz, and the satellite velocity is approximately 7 km/s. Substituting these values (and assuming high SNR values) yields a measurement error ranging from approximately 0.15 dB to 0.13 dB depending on the PRF. Hence total PIA uncertainty is estimated as:

$$\sigma_{PIA}^{uncer}[dB] = \sqrt{(\sigma_{gas}^{uncer})^2 + (\sigma_z)^2} \tag{13}$$

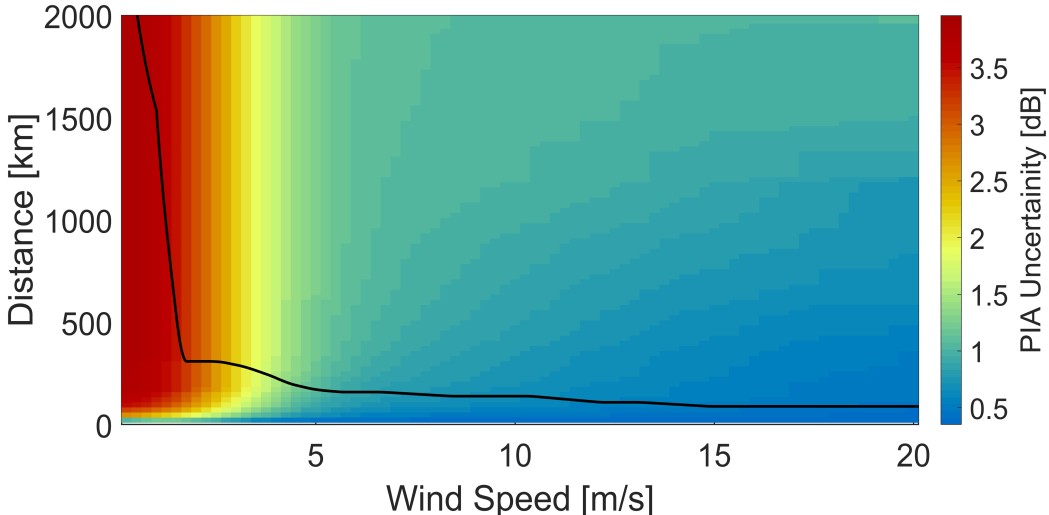

**Figure 1.** Lookup table of two-way PIA uncertainty associated with the clear-sky interpolation method, shown as a function of wind speed at the cloudy profile and distance to the calibration points. The uncertainty is derived using an ensemble of clear-sky oceanic profiles over the June 2024–June 2025 period. For each profile location $x$, the NRCS is estimated using all other clear-sky profiles as calibration points. The difference between the estimated and measured NRCS at $x$ defines the residual. These residuals are then binned by wind speed and distance to the calibration points, and the uncertainty in each bin is quantified as the standard deviation of the residuals as in Eq. (9). The solid black contour delineates the transition boundary beyond which the model-driven method, Eq. (5), yields lower PIA uncertainty compared to the clear-sky interpolation approach.

This approach is similar to the one already adopted for CloudSat but with two major differences.




1. In CloudSat, the interpolation method is typically applied only when clear-sky pixels are immediately adjacent to the cloudy pixel of interest, usually within a window of thirty surrounding profiles, which corresponds to $\approx 30$ km (2C-PRECIP-COLUMN Product Description, 2018). In contrast, the method used here allows interpolation even when the calibration points are $\approx 200$ km to $\approx 100$ km from the cloudy pixel in wind speed conditions between 4 and 15 m/s (Fig. 1). This is possible because the variability in the gas absorption profile and in the $\sigma_{0e}$ due to the modulation of the atmospheric (temperature and relative humidity profile) and of the surface (SST and wind) properties, respectively, is accounted for in Eq. (6).

2. Each calibration point used in the PIA estimation is weighted based not only on it's distance from the point of interest but also on the potential uncertainty associated with wind speed at that location (see Sect. 2.4).

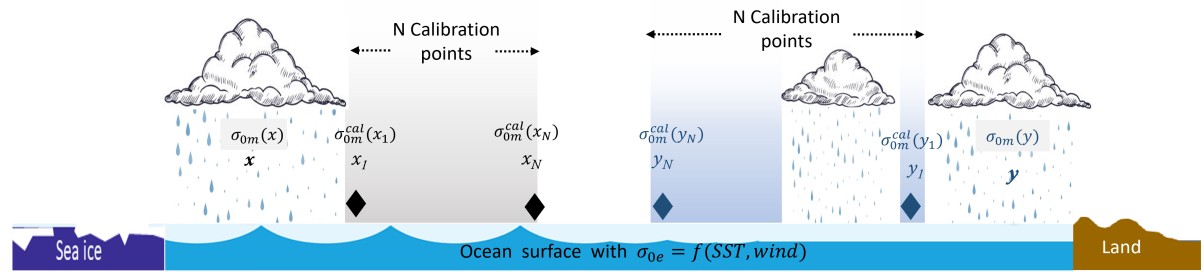

**Figure 2.** Schematic representation of PIA estimation methodology described in Sect. 2. The cloudy profiles of interest (located at positions $x$ and $y$) are bordered by $N$ clear-sky calibration points ($x_1, \ldots, x_N$ and $y_1, \ldots, y_N$). The measured normalized radar cross Section (NRCS) at a cloudy location is denoted as $\sigma_{0m}(x)$ and $\sigma_{0m}(y)$ , while $\sigma_{0m}^{cal}(x_i)$ and $\sigma_{0m}^{cal}(y_i)$ refer to NRCS values at the calibration points. The effective surface backscatter under clear conditions, $\sigma_{0e}$, is modeled as a function of sea surface temperature (SST) and surface wind speed: $\sigma_{0e} = f(SST, wind)$. The domain is bounded by sea ice on the left and land on the right, with calibration only valid over ice-free open ocean.

## 2.1 Gas attenuation modelling

In high-frequency radars operating at 94 GHz, microwave radiation is significantly absorbed by atmospheric gases, primarily water vapor and oxygen, as the radar signal propagates through the atmosphere. This absorption contributes to the total path-integrated attenuation (PIA) and must be accounted for in retrieval algorithms. In the EarthCARE C-PRO FMR dataset (Kollias et al., 2023), gaseous attenuation is estimated using the Rosenkranz absorption model (Rosenkranz, 1998), with temperature and moisture profiles provided by X-MET, matched to the radar observations.

## 2.2 Derivation of normalized surface backscattering cross section

The normalized surface back-scattering cross section, $\sigma_{0m}$ (first term on the right-hand side of Eq. (3)), is derived from the received reflectivity profile by identifying $Z_{clutter}(r_{surf})$, which is the reflectivity corresponding to the surface (Kanemaru





et al., 2020), (EarthCARE CPRL1b ATBD, 2017). The expression used is:

$$\sigma_{0m} = \frac{\pi^5 |K_w|^2}{\lambda^4} \frac{c\tau_p}{2} L_p \, Z_{clutter}(r_{surf}) \tag{14}$$

where $K_w$ is derived from the refractive index of water at 3 mm-wavelengths ($|K_w|^2$ assumed equal to 0.75) and $L_p$ is a peak loss factor (that can be computed from calibration), that accounts for the receiver transfer function and for the fact that the pulse shape is not a perfect top hat.

The EarthCARE CPR has a vertical range sampling of 100 meters ($\Delta r$), meaning the actual surface height ($r_{surf}$) often falls between two discrete range bins and is generally missed. The "surface_bin_number" ($n_{surf}$) variable in CPR L1B data represents the range bin index where the peak reflectivity was detected and the corresponding height ($r(n_{surf})$) is the sampled range closest to the surface range. The NRCS reported in CPR L1B data corresponds to this "surface_bin_number"and therefore must be corrected for potential peak loss due to the coarse vertical resolution. For accurate estimation of surface height,

gaussian fitting is performed on the surface reflectivity peak in the CPR L1B data (EarthCARE JAXA L2 ATBD, May 2024). The variable "surface_bin_fraction" ($f_{surf}$) represents the offset between actual surface range ($r_{surf}$) obtained by the fitting and the closest sampled range ($r(n_{surf})$) expressed as a fraction of the bin size. In EarthCARE the $f_{surf}$ ranges from -0.5 to 0.5 where negative $f_{surf}$ values indicates that the actual surface range is above the closest sampled range and viceversa. The actual surface height can be calculated as:

$$r_{surf} = r(n_{surf}) - f_{surf} \, \Delta r \tag{15}$$

In computation of $\sigma_{0m}$ with the Eq. (14), the reflectivity at bin $n_{surf}$, $Z_{clutter}(r(n_{surf}))$, is used (reported in CPR L1B data) and a correction is applied for the peak loss. To compute the correction term for $\sigma_{0m}$, a large ensemble of clear-sky profiles over ice-free open ocean was collected. For each profile, actual surface height is estimated using Eq. (15) and profiles were aligned relative to the distance from this surface detected height so that if the radar samples actual surface height ($r_{surf}$),

the peak reflectivity will be at 0 m. These profiles were then averaged to derive best point-target response (PTR) function (Coppola et al., 2025). In the derived PTR, the reflectivity loss within +50 m is 0.48 dB and for -50 m is 0.138 dB.

Substituting the constants into Eq. 14 indicates that a surface reflectivity of approximately 29.65 dBZ produces a $\sigma_{0m}$ of 0 dB (when assuming $L_p = 1$). So the Eq. 14 can be re-written as:

$$\sigma_{0m}(dB) = Z_{clutter}(r_{surf})(dBZ) - 29.65 \tag{16}$$

The peak loss correction $L_r^{dB}(f_{surf})$ is expressed as:

$$L_r^{dB}(f_{surf}) = \begin{cases} -0.965 f_{surf}, & -0.5 \le f_{surf} \le 0 \\ 0.276 f_{surf}, & 0 < f_{surf} \le 0.5 \end{cases}$$

and, $\sigma_{0m}$ is corrected for peak loss as:

$$\sigma_{0m}^{corr}(dB) = \sigma_{0m}(dB) + L_r^{dB}(f_{surf}) \tag{17}$$





The NRCS measurement is available unless the surface signal is completely attenuated by heavy precipitation or thick cloud layers. The minimum detectable reflectivity of EarthCARE CPR is approximately -35 dBZ. Over ocean surfaces with wind speeds of 6-8 m/s, the most frequently observed $\sigma_{0e}$ values range between 10 and 15 dB (Fig. 4). Using Eq. (16), a surface reflectivity of -35 dBZ corresponds to a measured $\sigma_{0m}$ of -64.65 dB. Assuming $\sigma_{0\,e}$ of 10 dB, the maximum detectable PIA, limited by the radar's sensitivity, is approximately 74.65 dB and if PIA were any greater, then the surface signal would fall below the radar's detection threshold.

## 2.3 Selection of calibration points

In the PIA estimation methodology proposed here, clear-sky and thin ice cloud only profiles are used as calibration points for interpolating the NRCS over cloudy regions. Therefore, accurate selection of these calibration points is critical for ensuring reliable PIA estimates. Currently, calibration points are identified exclusively using radar-based products, specifically the significant detection mask, or "profile_class", from the Level 2 C-PRO FMR dataset (Kollias et al., 2023). A profile flagged as clear by the mask is confirmed as a calibration point only if, within a 10 km along-track segment centered on it, at least six other profiles are also classified in the same way, and the standard deviation of their NRCS is less than 0.3 dB. This threshold is derived from global climatological statistics of standard deviation of NRCS, computed over 10 km along-track segments that contain at least six clear-sky profiles, consistent with the calibration point selection criteria. The global climatological analysis of NRCS standard deviation reveals that the most frequently occurring values lie between 0.2 dB and 0.3 dB, and higher standard deviation is observed near coastal regions. The standard deviation threshold helps to avoids selecting isolated clear-sky profiles that may be incorrectly flagged due to noise or retrieval errors and guarantees selected profiles represent typical, stable clear-sky surface conditions. Figure 3 shows the global distribution of calibration point fraction, defined as the ratio of the number of calibration points to the total number of profiles within each $1° \times 1°$ grid cell. The black dashed contour lines in Fig 3 indicate the fraction of profiles where clear-sky interpolation method was applied, relative to the total number of profiles for which PIA was estimated. A general trend can be observed with the fraction of profiles using clear-sky interpolation declining as the availability of calibration points decreases.





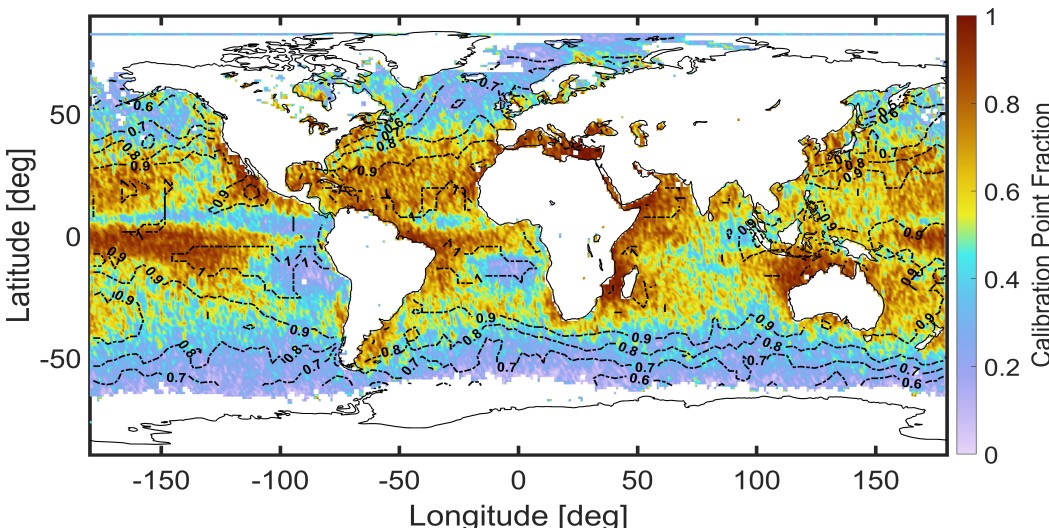

**Figure 3.** Global distribution of the calibration point fraction, defined as the ratio of valid calibration points to the total number of radar profiles within each $1° \times 1°$ grid cell. A profile is considered a valid calibration point if it is flagged as clear by the "profile_class" mask in the Level 2 C-PRO FMR dataset and is located within a 10 km along-track segment that contains at least six other clear profiles, with a standard deviation of measured NRCS below 0.3 dB. Black dashed contour lines indicate the fraction of profiles where the clear-sky interpolation method is applied, highlighting regions where this approach is frequently used for PIA estimation.

## 2.4  $\sigma_{0e}$ modelling

The effective normalized radar cross Section ($\sigma_{0e}$) over the ocean surface is a key parameter in the estimation of PIA, as it captures the expected variability of surface backscatter as a function of radar viewing geometry, surface wind speed, and

SST. The dependence on wind speed arises from wind-driven waves that increase surface roughness, which is characterized by the effective mean square slope (MSS) of the ocean surface. According to quasi-specular scattering theory, $\sigma_{0e}$ is inversely proportional to the square of MSS. The MSS itself is primarily a function of wind speed and has been empirically related to wind velocity through models developed by Cox and Munk (1954), Wu (1972, 1990), and Freilich and Vanhoff (2003)(Li et al., 2005). Additionally, SST influences $\sigma_{0e}$ through its effect on the refractive index of seawater, which alters the Fresnel reflection

coefficients. The $\sigma_{0e}$ can either be estimated using a geophysical model using wind and SST measurements from ECMWF data or can be estimated from measured NCRS at clear-sky conditions by correcting for gaseous attenuation. Figure 4 illustrates the variation of measured $\sigma_{0e}$ with wind speed, based on clear-sky profiles observed by the EarthCARE CPR between June 2024 and February 2025. Clear-sky conditions were identified using the "profile_class" variable from the Level 2 C-PRO FMR product (Kollias et al., 2023), and the analysis was limited to ice-free oceanic regions as indicated by the ECMWF auxiliary

sea-ice mask. The figure demonstrates a clear dependence of $\sigma_{0e}$ on wind speed, with mean values ranging from approximately 5 dB to 18 dB. The error bars represents the standard deviation, capturing the variability of data. Notably, greater variability




in $\sigma_{0e}$ is observed under low wind conditions, where surface roughness is minimal and NCRS is more sensitive to small-scale variations(Haynes et al., 2009).

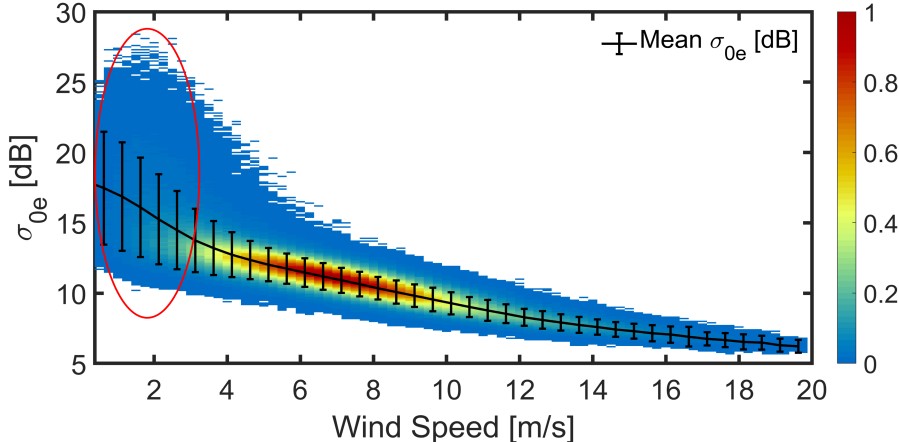

**Figure 4.** Distribution of measured $\sigma_{0e}$ derived from EarthCARE NRCS observations under clear-sky conditions, corrected for gaseous attenuation, over the period June 2024-June 2025, shown as a function of wind speed. The black curve represents the mean $\sigma_{0e}$ at each wind speed, with error bars indicating the standard deviation. The red circle highlights the low wind speed regime, where $\sigma_{0e}$ exhibits greater variability and a higher standard deviation.

Figure 5 shows the variability of $\sigma_{0e}$, expressed as the standard deviation computed over 10 km along-track segments under

clear-sky conditions. The calculation is performed only when at least six clear-sky pixels are available within each segment. The black curve indicates the median of the resulting distribution and error bars represent 25 and 75th percentiles. In higher wind regimes, the median standard deviation is approximately 0.25 dB, reflecting the expected reflectivity measurement uncertainty associated with signal-to-noise ratio (SNR). As wind speed decreases, the median standard deviation increases, reaching up to 1 dB in low-wind conditions. This reflects the increased sensitivity of the $\sigma_{0e}$ to small-scale surface variations under calm

ocean conditions.



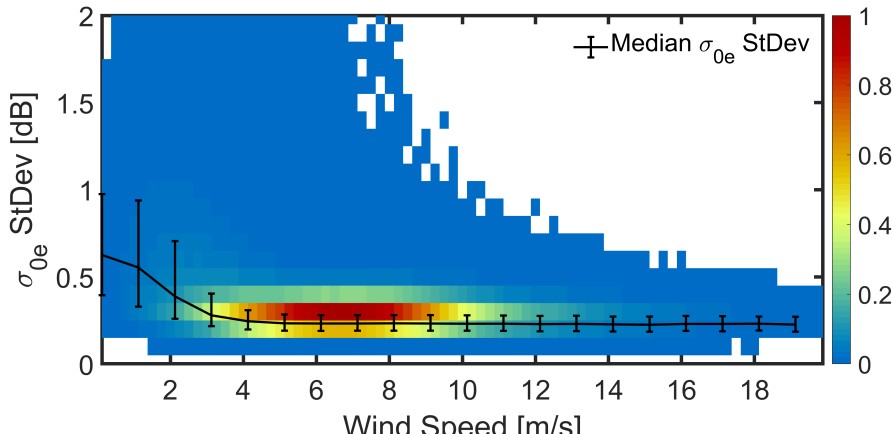

**Figure 5.** Variability of measured $\sigma_{0e}$ from EarthCARE over the period June 2024-June 2025, expressed as the standard deviation within 10 km along-track segments. Only segments containing at least six clear-sky pixels are included. The black curve represents the median of the distribution, and the error bars indicate the 25th and 75th percentiles.

The measured $\sigma_{0e}$ should align with $\sigma_{0e}$, estimated from geophysical models, and this is examined in Fig. 6.

Figure 6 shows a comparison between the mean $\sigma_{0e}$ from EarthCARE measurements and different geophysical model estimates. The $\sigma_{0e}$ estimated using the Cox and Munk (1954) empirical relationship provides the best agreement with the measured mean $\sigma_{0e}$, showing minimal bias across most wind speed ranges, except at very low wind speeds below 2 m/s. To reduce po-255 tential biases associated with geophysical model based estimates across different wind speed regimes, the current PIA retrieval algorithm in the EarthCARE Level 2 C-PRO FMR product derives the $\sigma_{0e}$ from a climatologically constructed look-up table. Figure 7 presents the look-up table of $\sigma_{0e}$ as a function of wind speed and sea surface temperature (SST), derived from EarthCARE observations over the period June 2024 to June 2025.



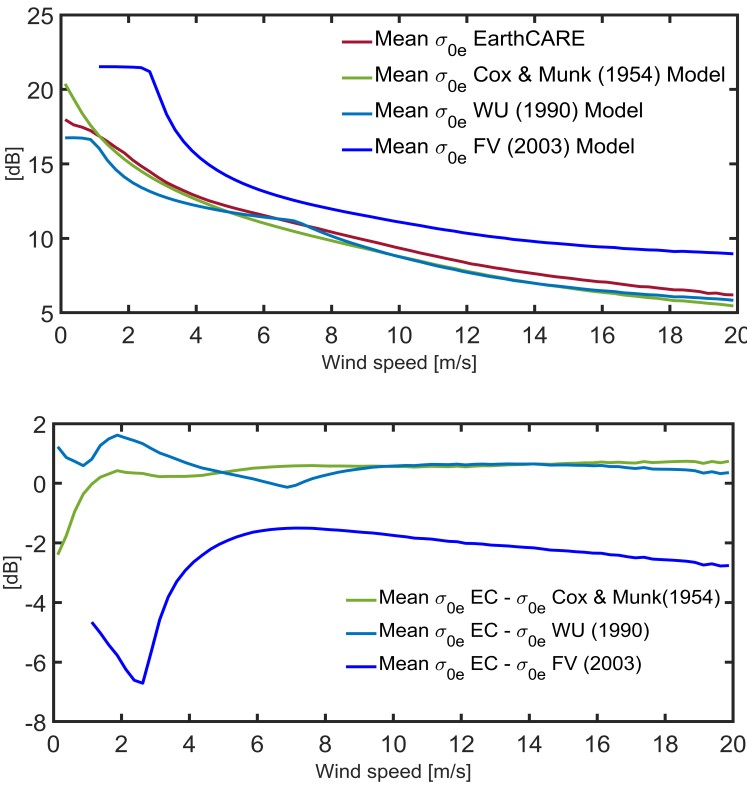

**Figure 6.** Comparison of mean $\sigma_{0e}$ from EarthCARE with estimates from various geophysical models. The top panel shows the mean $\sigma_{0e}$ measured by EarthCARE across wind speed bins, alongside the corresponding mean $\sigma_{0e}$ values from different geophysical models. The bottom panel displays the differences between the EarthCARE measurements and each model estimate.

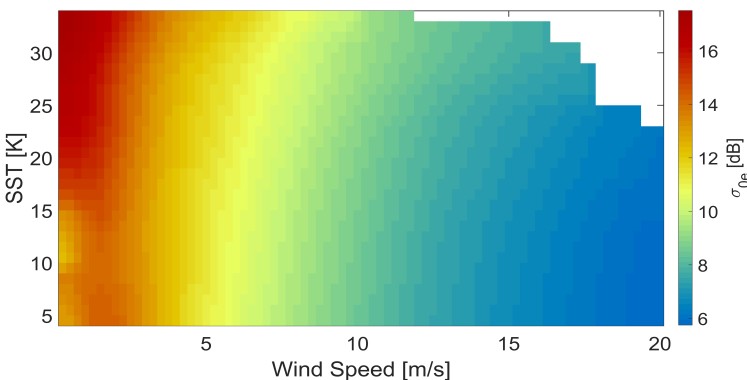

**Figure 7.** Look-up table of the effective surface backscattering cross section ($\sigma_{0e}$) derived from EarthCARE measurements collected between June 2024 and June 2025, shown as a function of sea surface temperature (SST) and wind speed. The table is constructed by averaging clear-sky NRCS observations corrected for gaseous attenuation within discrete SST and wind speed bins. Results indicate that $\sigma_{0e}$ exhibits significantly greater variability with respect to wind speed than SST.





In the EarthCARE analysis, wind data are obtained from ECMWF reanalysis. Given the high variability of the $\sigma_{0e}$ in low-
wind conditions, coupled with potential errors in ECMWF wind speed estimates, PIA retrievals particularly in the regions
highlighted by red circles in Fig. 4 are expected to exhibit increased uncertainty and reduced reliability. This increased uncer-
tainty is also reflected in the PIA uncertainty look-up table (Fig. 1), resulting in higher reported PIA uncertainty for cloudy
profiles occurring at low wind speeds.

## 3   EarthCARE case studies

To demonstrate the performance of the proposed PIA estimation methodology under varying atmospheric and surface condi-
tions, several case studies using EarthCARE observations are presented in Fig. 8-10. In each case, the first panel displays the
vertical profiles of the radar reflectivity factor as a function of the along track distance with the calibration points selected
based on the criteria outlined in Sect. 2.3 and shaded in grey. The second panel shows the measured NRCS ($\sigma_{0m}$), which may
be attenuated by hydrometeors, alongside the estimated gas-only NRCS ($\sigma_0^{gas}$). The $\sigma_0^{gas}$ and corresponding PIA are com-
puted using five clear-sky calibration points as defined by Eq. (7). The third panel presents the resulting PIA estimates, with
shaded regions indicating profiles where negative PIA values are obtained. The fourth panel shows the total PIA uncertainty
estimate, calculated using Eq. (13), which includes both the uncertainty from the PIA uncertainty look-up table (Fig. 1) and a
fixed contribution of 0.15 dB from inherent measurement noise, corresponding to a PRF of 6100 Hz. The shading in the panel
represents calibration points where PIA is not estimated.

Figure 8 depicts a scene of scattered cumulus clouds over the Southern Ocean. Note that, thanks to the sharp EarthCARE
point target response (Burns et al., 2016; Lamer et al., 2020; Coppola et al., 2025), the profiles of radar reflectivity are not
contaminated by clutter down to 500 m. In this case, numerous calibration points are situated close to the cloudy profiles,
allowing for high-confidence PIA estimates with relatively low uncertainty. The farthest calibration point is approximately
50 km away, and the maximum PIA uncertainty is 0.4 dB which is presented in fourth panel. Wind speeds in this scene range
between 3.5 and 8.8 m/s.




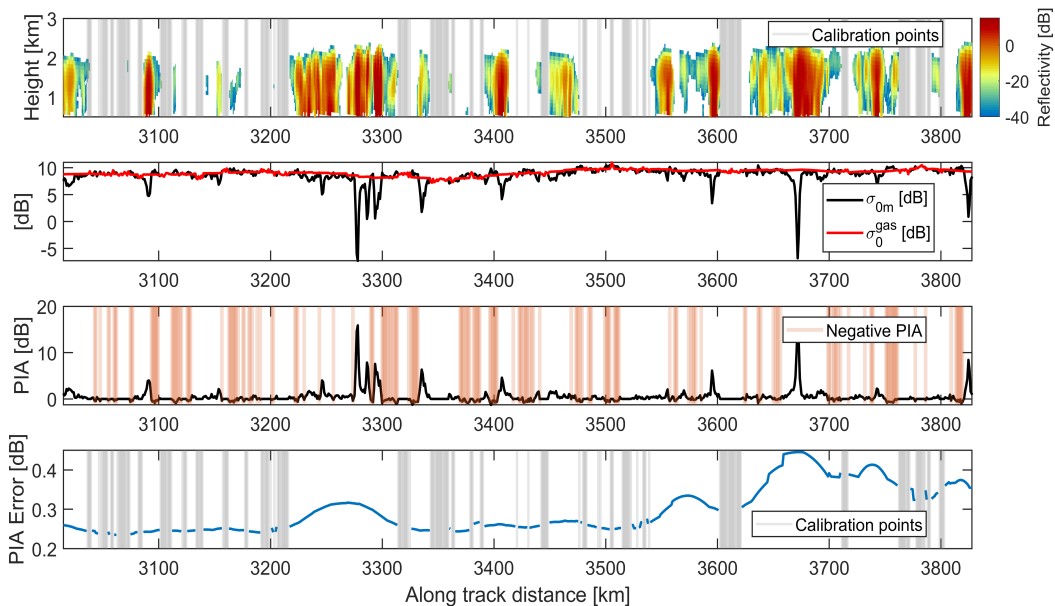

**Figure 8.** EarthCARE Case 1: Scattered shallow cumulus clouds observed off the southeastern coast of Africa. The first panel highlights the selected calibration points (shaded areas). The second panel compares the measured NRCS ($\sigma_{0m}$) with the estimated clear-sky NRCS ($\sigma_0^{gas}$), representing the expected NRCS in presence of gas only, derived using Eq. (7). The third panel presents the resulting PIA estimates, with shaded regions indicating profiles where the estimated PIA is negative. The fourth panel presents the error in PIA estimate derived based on the PIA uncertainty look-up table (Fig. 1).

Figures 9 and 10 depict extensive, continuous cloud systems with few or no nearby calibration points. Figure 9 shows a persistent stratocumulus deck over the southeastern Atlantic Ocean, off the southwestern coast of Africa. These clouds are typically shallow with flat tops and are capped by a temperature inversion. Although the resulting PIA values remain relatively low (generally below 2-3 dB), accurate estimation is essential for reliable rainfall retrievals. In this case, for profiles in the middle of the precipitating system, the farthest calibration point can be located approximately 480 km away, corresponding to a maximum PIA uncertainty of 0.8 dB as shown in the fourth panel. Wind speeds in the scene range from 7 to 11 m/s, and the cloud deck extends roughly 1170 km in length. Figure 10 illustrates a widespread stratiform cloud system over the southeastern Atlantic Ocean near the western coast of Africa. The cloud cover stretches nearly 930 km, with limited or no nearby calibration points. The farthest calibration point is about 340 km away, resulting in a maximum estimated PIA uncertainty of 0.45 dB which is represented in the fourth panel. Wind speeds in this region range from 7 to 10.5 m/s.



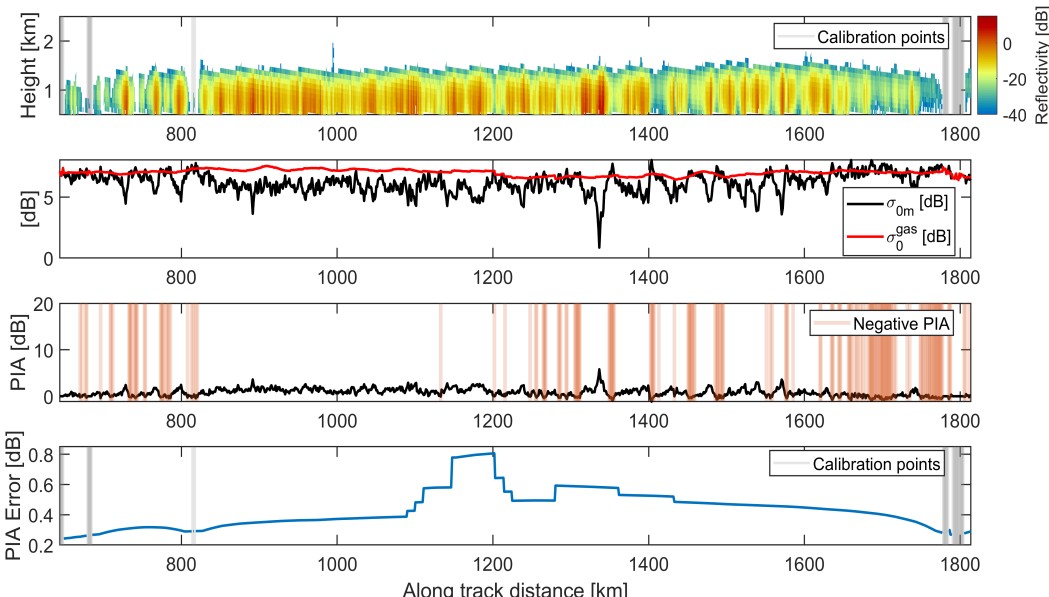

**Figure 9.** EarthCARE Case 2: Stratocumulus case seen over the southeastern Atlantic Ocean, off the southwestern coast of Africa. The first panel highlights the selected calibration points (shaded areas). The second panel compares the measured NRCS ($\sigma_{0m}$) with the estimated clear-sky NRCS ($\sigma_0^{gas}$), representing the expected NRCS in presence of gaseous attenuation only, derived using Eq. (7). The third panel presents the resulting PIA estimates, with shaded regions indicating profiles where the estimated PIA is negative. The fourth panel presents the error in PIA estimate derived based on the PIA uncertainty look-up table (Fig. 1).



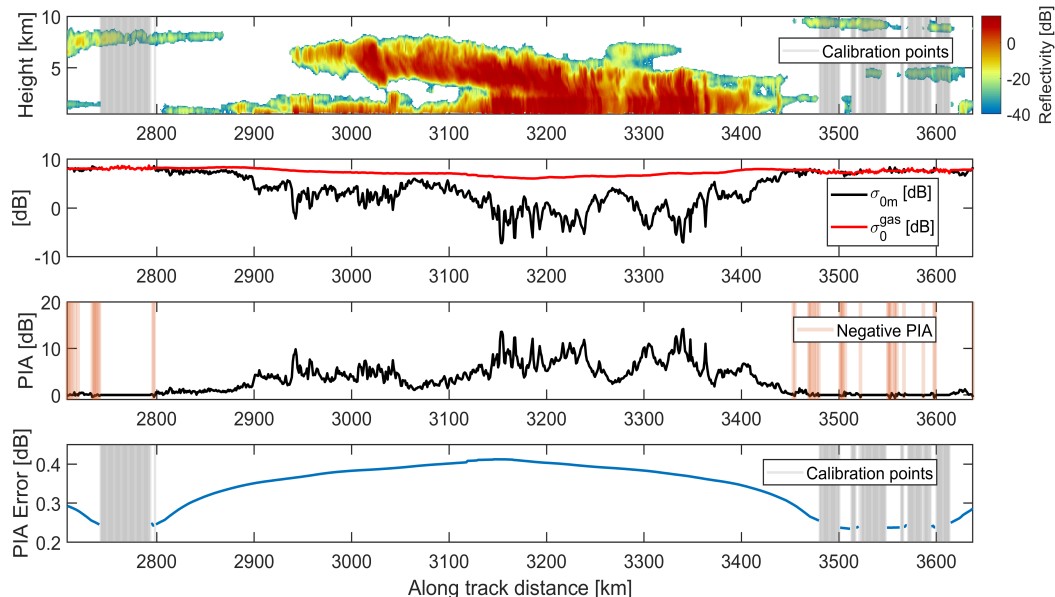

**Figure 10.** EarthCARE Case 3: Stratiform cloud seen over southeastern Atlantic Ocean near the western coast of Africa. The first panel highlights the selected clear-sky calibration points (shaded). The first panel highlights the selected calibration points (shaded areas). The second panel compares the measured NRCS ($\sigma_{0m}$) with the estimated clear-sky NRCS ($\sigma_0^{gas}$), representing the expected NRCS in the presence of gaseous attenuation only, derived using Eq. (7). The third panel presents the resulting PIA estimates, with shaded regions indicating profiles where the estimated PIA is negative. The fourth panel presents the error in PIA estimate derived based on the PIA uncertainty look-up table (Fig. 1).

In all the cases above, the negative PIA values are small, typically fractions of a dB. These negative estimates arise from the noisiness in the measured $\sigma_{0m}$ associated to the fluctuations of the backscattering returns and from the uncertainties associated in $\sigma_0^{gas}$ (e.g. associated to the ECMWF reanalysis wind speed and SST used as inputs).

These diverse case studies highlight the flexibility and robustness of the proposed PIA retrieval approach across different
cloud morphologies, calibration point availability, and wind conditions.

## 4   CloudSat PIA testbed

As briefly discussed in Sect. 1, CloudSat employs a hybrid approach to estimate PIA, combining two complementary methods similar to the one proposed in this study. The first approach, referred to as the Wind/SST method, estimates the NRCS at cloudy region in absence of hydrometeor and presence of gaseous attenuation ($\sigma_0^{gas}$) as a function of surface wind speed and SST using geophysical models (Li et al., 2005) and second one is interpolation-based approach, where clear-sky profiles
surrounding cloudy profile are used to estimate the $\sigma_0^{gas}$. In the interpolation-based method, a search is performed within 30 profiles (approximately 30 km) surrounding cloudy profile for clear-sky conditions. If at least five clear profiles are found, a weighted average of their observed NRCS is computed, with weights based on the distance of each clear profile to the cloudy



profile (2C-PRECIP-COLUMN Product Description, 2018). If the minimum requirement of five clear-sky profiles is not met,
the Wind/SST method is used instead.

Figure 11 presents a case study from 02 January 2008, using CloudSat observations. The PIA estimation methodology
proposed in this study is also applied to this case for direct comparison with CloudSat method. CloudSat provides estimates of
the unfiltered PIA (i.e., without discarding negative values), along with the measured NRCS ($\sigma_{0m}$). Therefore the estimate of
NRCS at cloudy region in presence of gas only ($\sigma_0^{gas}$) for the CloudSat products can be obtained by just summing the PIA and
the measured $\sigma_{0m}$.

The second panel of Fig. 11 shows the measured NRCS ($\sigma_{0m}$), the $\sigma_0^{gas}$ based on CloudSat method ($\sigma_0^{gas}$ CloudSat), and
the estimated $\sigma_0^{gas}$ based on our methodology proposed for EarthCARE ($\sigma_0^{gas}$ EarthCARE). Abrupt jumps up to nearly 1 dB
are observed in the CloudSat-derived $\sigma_0^{gas}$, particularly at transition points between the two estimation methods. These dis-
continuities are marked by red circles in the second panel of Fig. 11. The variable "Diagnostic_PIA_method" in the CloudSat
2C-PRECIP-COLUMN product indicates which method is used at each profile, which is represented by two shading in the sec-
ond panel of Fig. 11. Blue shading represents the interpolation-based method, while grey shading corresponds to the Wind/SST
method.

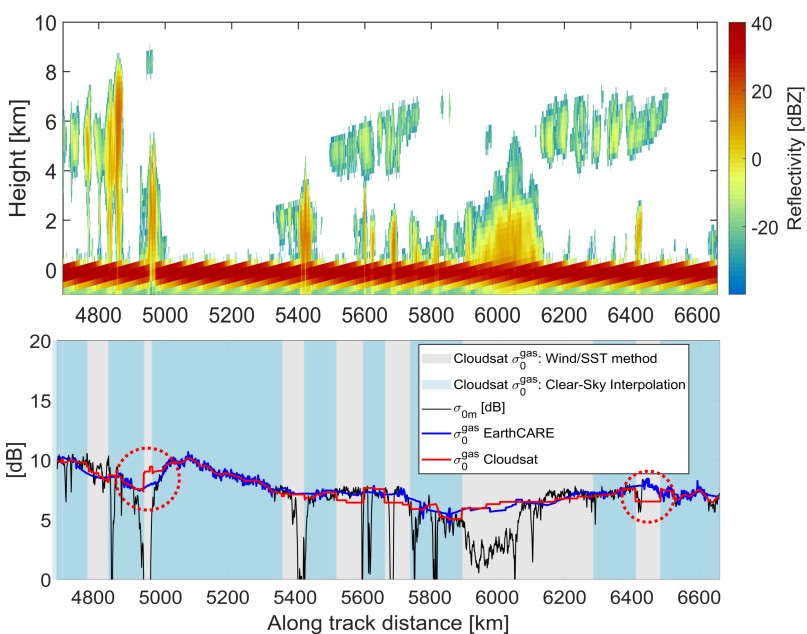

**Figure 11.** CloudSat Case Study. Top panel: vertical reflectivity profile as a function of the along track distance. Bottom panel: the measured
normalized radar cross Section (NRCS), denoted as $\sigma_{0m}$ (black curve), along with the estimated clear-sky NRCS ($\sigma_0^{gas}$) within cloudy
regions, derived using the proposed EarthCARE method (blue curve) and the CloudSat-based estimate (red curve). The red circle highlights
the jump in $\sigma_0^{gas}$ in CloudSat estimate. The grey and blue shading in second panel represent the two estimation methods employed in CloudSat
methodology, which are Wind/SST method which utilizes geophysical model and clear-sky interpolation method. The jumps present in $\sigma_0^{gas}$
CloudSat generally occur when there is switch in two methodologies.





The 30 km limit of CloudSat's clear-sky interpolation often leads to frequent switches to the Wind/SST-based method, causing nonphysical jumps in $\sigma_0^{gas}$ and PIA. In contrast, the EarthCARE approach allows interpolation over much longer
distances typically between 200 km to 100 km depending on the surface wind speed, significantly reducing such transitions and yielding smoother, more consistent estimates. Although method transitions may still introduce occasional discontinuities in the EarthCARE estimates, their frequency is markedly lower compared to the CloudSat approach.

## 4.1 Statistical comparison with the CloudSat PIA estimates

To facilitate a direct comparison between the PIA estimation methodology implemented in EarthCARE and that used in Cloud-
Sat, the proposed method is applied to a four-month subset of CloudSat data, spanning January to April 2007. The effective normalized radar cross Section ($\sigma_{0e}$) is derived using a look-up table based on ECMWF wind speed and SST, generated over the entire CloudSat mission epoch spanning from 05 August 2006, to 16 December 2021. Clear-sky profiles are identified solely using radar-based products. In particular, the "CPR_Echo_Top" variable from the 2C-PRECIP-COLUMN product (2C-PRECIP-COLUMN Product Description, 2018) is used to distinguish between clear-sky and cloudy profiles. Calibration
points are selected according to the criteria detailed in Sect. 2.3. Figure 12 presents a statistical comparison of PIA estimates derived from the proposed method and those reported by CloudSat, considering only cloudy profiles on a global scale. The distributions are displayed on a logarithmic scale to better represent the range of occurrences. The results indicate that both methods produce similar statistical characteristics, with histograms peaking in the same range (0-1 dB) and exhibiting comparable widths, reflecting overall agreement. The EarthCARE method shows a slightly lower occurrence of small negative PIA
values (0 to -1 dB) and a marginally higher occurrence of small positive values (0-2 dB) compared to CloudSat. Additionally, while the EarthCARE approach yields a higher number of PIA estimates in the larger negative range (-2 to -5 dB), these cases remain relatively rare, resulting in a low associated probability density. Overall, the consistency in histogram shape and central tendency supports the validity of the EarthCARE PIA estimation methodology.





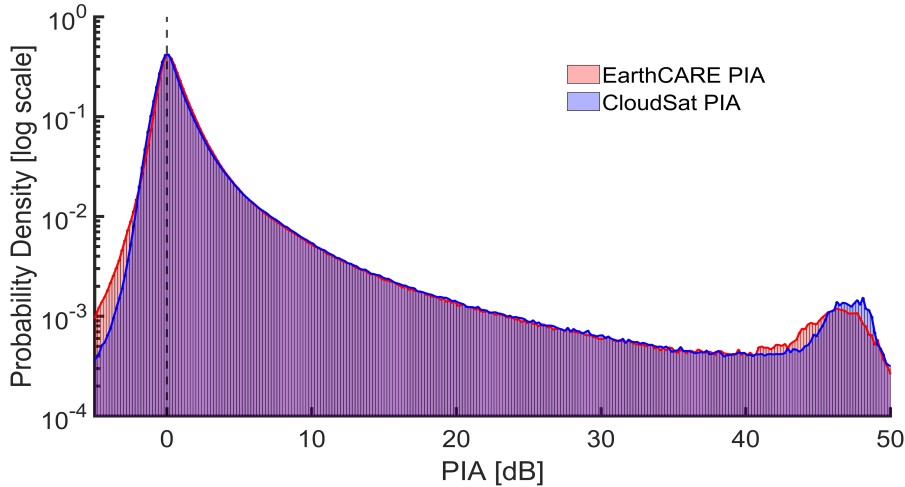

**Figure 12.** The global probability distributions of PIA estimates obtained from the EarthCARE retrieval methodology applied to CloudSat observations (January–April 2007) and from the PIA from CloudSat retrievals are compared on a logarithmic scale. The overlapping distributions demonstrate strong consistency in the statistical characteristics of the two retrieval approaches, supporting the robustness of the EarthCARE method when applied to CloudSat data.

As described in Sect 4, CloudSat employs a hybrid strategy for PIA estimation that combines clear-sky interpolation and the wind/SST-based method. The variable "Diagnostic_PIA_method" from the CloudSat 2C-PRECIP-COLUMN product (2C-PRECIP-COLUMN Product Description, 2018) indicates which retrieval method is applied to each profile. Leveraging this information, profiles are categorized based on the applied PIA estimation method, enabling a more granular comparison. Within each category, the CloudSat PIA estimates are compared against those produced by the proposed EarthCARE methodology, allowing for a detailed assessment of consistency and potential differences across retrieval strategies.

Figure 13 presents the distribution of differences between PIA estimates from those derived using the EarthCARE method and Cloudsat method, categorized by the PIA retrieval approach applied in CloudSat. For profiles where the clear-sky interpolation method is implemented in Cloudsat, the differences are generally minor, with the histogram centered around 0 dB and most values falling within the ±0.5 dB range. It indicates that, for profiles where CloudSat used clear-sky interpolation, the PIA estimates from both CloudSat and the EarthCARE method are in close agreement. In contrast, for profiles retrieved using the wind/SST method, the discrepancies are significantly larger, with differences reaching up to approximately ±2 dB, indicating greater divergence between the two methods in these cases. Here, since the difference is calculated as PIA (EarthCARE) minus PIA (CloudSat), the distribution skews more positive, indicating that EarthCARE's PIA estimates tend to be higher than CloudSat's for these profiles. This difference arises primarily because the EarthCARE methodology relies less frequently on the Wind/SST approach than CloudSat. Specifically, in the EarthCARE implementation, clear-sky interpolation is applied to 77.14% of the profiles, with the Wind/SST method used in only 22.86% of cases. Conversely, the CloudSat method applies clear-sky interpolation to just 33.91% of profiles, relying on the Wind/SST method for the remaining 66.09%.




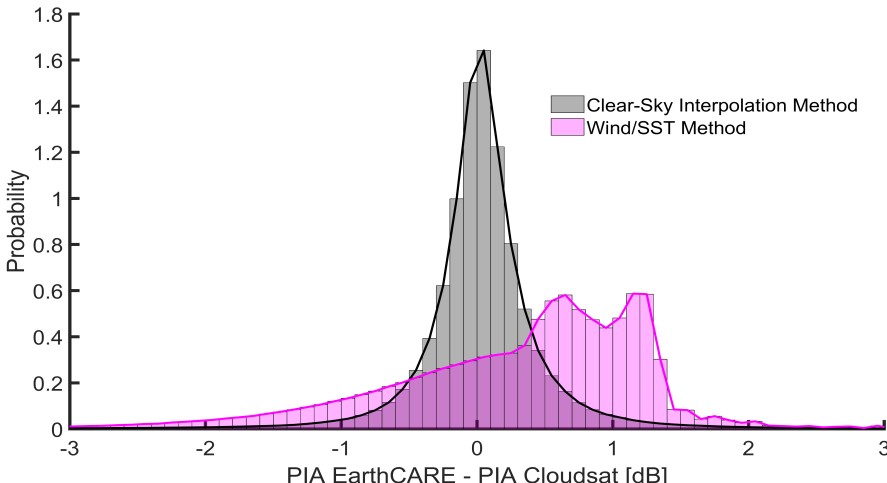

**Figure 13.** Probability distributions of the differences between PIA estimates from EarthCARE and CloudSat for cloudy profiles. The gray histogram represents cases where CloudSat applies the clear-sky interpolation method, showing a narrow distribution centered around 0 dB. The purple histogram corresponds to profiles where the wind/SST method is used in CloudSat, exhibiting a broader distribution with larger discrepancies.

These statistical comparison indicates that the PIA estimates derived from the proposed EarthCARE method are largely consistent with those from CloudSat. Moreover, the newly proposed EarthCARE method demonstrates improvement, exhibiting a reduced occurrence of negative PIA values.

## 5 Summary

This study presents a robust methodology for estimating path-integrated attenuation (PIA) over oceanic regions, which is currently under implementation and will be incorporated into the PIA estimation component of the EarthCARE CPR Level 2A C-PRO product. The approach is specifically designed to be resilient to potential radar calibration biases, such as those that may arise during the early phases of the mission, thereby enhancing the reliability of attenuation-based retrievals under non-ideal instrument conditions. It combines two complementary approaches: a clear-sky interpolation technique and a model-driven (wind/SST) method. The clear-sky interpolation method estimates PIA at a cloudy profile by leveraging surrounding calibration points selected based in criteria described in Sect. 2.3, as defined in Eq. (7). Importantly, the clear-sky interpolation method, as described in Eq. (6), estimates PIA by computing the difference between the measured normalized radar cross section (NRCS) (or effective surface backscattering cross section corrected for gaseous attenuation) at the cloudy profile and that at surrounding clear-sky calibration points, rather than relying on their absolute values. The method uses multiple calibration points, optimally weighted based on their distance from the cloudy profile and the surface wind speed at the cloudy profile, so that the nearest calibration points are weighted higher and the PIA estimate at a cloudy profile at low wind conditions report larger uncertainty.



In situations where suitable clear-sky calibration points are not available within a distance that permits accurate interpolation, the retrieval defaults to a model-based approach, as described in Eq. (5). The model-based method estimates PIA using the effective normalized radar cross section ($\sigma_{0e}$), derived from climatology-based look-up table that relates $\sigma_{0e}$ to surface wind speed and sea surface temperature (SST), based on collocated ECMWF data. The hybrid method can be applied when the radar is well calibrated.

The performance of the EarthCARE method was evaluated by applying it to CloudSat observations over four months and by comparing the resulting PIA estimates to those reported in CloudSat's 2C-PRECIP-COLUMN product (2C-PRECIP-COLUMN Product Description, 2018). CloudSat uses a similar hybrid strategy, choosing between clear-sky interpolation and a wind/SST-based approach depending on the availability of nearby clear-sky profiles. However, CloudSat applies clear-sky interpolation only within a 30 km, while the EarthCARE approach allows interpolation from calibration points located 100 to 200 km away from cloudy profile, depending on the surface wind speed. This extended interpolation capability reduces the number of transitions between estimation methods and improves the spatial consistency of the retrieved PIA.

A detailed case study and global statistical analysis confirm the effectiveness of the proposed EarthCARE methodology. For profiles where CloudSat applies clear-sky interpolation, both methods yield highly consistent PIA values, with most differences falling within $\pm 0.5$ dB. In contrast, for profiles where CloudSat switches to the wind/SST method, larger discrepancies emerge, with differences occasionally reaching up to $\pm 2$ dB. This is partly because CloudSat applies the wind/SST method more frequently, over 66% of profiles globally compared to 23% in the EarthCARE method, which maintains a higher reliance on clear-sky interpolation. In general EarthCARE method provided PIA estimate with marginally lesser negative PIA estimates and higher occurrence of positive PIA estimates.

Overall, the proposed retrieval scheme demonstrates strong agreement with CloudSat's established method. In future work, other EarthCARE instruments beyond the radar, such as the Multi-Spectral Imager (MSI) and the Atmospheric Lidar (ATLID), can be leveraged in order to better identify clear-sky profiles, to validate the PIA estimates and to improve estimates of the LWP product (Lebsock et al., 2022). A brightness temperature product for the CPR, envisaged to be developed in the next months similarly to what was done for CloudSat (Battaglia and Panegrossi, 2020), could help in better constraining such product as well.

*Author contributions.* AB conceived the idea and provided overall supervision. SS contributed to methodology development and algorithm implementation, conducted the analysis, and drafted the manuscript and figures. BPT played a key role in refining the PIA parameterization and peak loss correction, supplying essential data, algorithm development and implementation of the algorithm on EarthCARE data, and reviewing the manuscript. PK supported the methodology and validation framework, contributed to manuscript revisions, and provided supervisory guidance. All authors reviewed and edited the manuscript, provided critical feedback, and helped shape the research and analysis.

*Competing interests.* At least one of the (co-)authors is a member of the editorial board of Atmospheric Measurement Techniques



*Acknowledgements.* The work conducted by Susmitha Sasikumar was undertaken as part of her PhD program at Politecnico di Torino and

is supported by the PNRR-NGEU project, which has received funding from the Italian Ministry of University and Research (MUR) under Ministerial Decree No. 118/2023. PK, and BPT were supported by the European Space Agency (ESA) under the EarthCARE Data Innovation and Science Cluster (DISC) project (AO/1-12009/24/I-NS). PK is also supported by the National Aeronautics and Space Administration (NASA) under the Atmospheric Observing System (AOS) project (Contract number: 80NSSC23M0113).



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
