# Peer review of "The Estimation of Path Integrated Attenuation for the EarthCARE Cloud Profiling Radar"

_EGUsphere, 2025_

## Referee Comment (RC2)

The authors describe a method for estimating the path integrated attenuation due to hydrometeors in an observed profile. The method uses two techniques, the first which uses measurements of surface backscatter in clear-sky profiles in the vicinity of the observed profile, and the second which used a scattering model to estimate the surface backscatter at the location of the observed profile. While CloudSat's 2C-RAIN-PROFILE product adopted a similar method, in this method the clear-sky profiles are allowed to be much further from the observed profile (~30 km for CloudSat versus 100-200 km for this method).

The paper is useful to document the methods used for EarthCARE products and so is likely appropriate for an EarthCARE special issue. I have one primary concern and that is that I think the uncertainty for the retrieved PIA has been estimated improperly. This estimation is described in lines 109 to 143.

In their approach, the authors compile a "large set" of clear-sky profiles. For a selected profile, differences described in equation (8) are computed against every other profile in the dataset and then decomposed by wind speed and separation distance. For each wind speed and distance bin, the standard deviation of the differences is taken, computing over all of the data points in the bin.

There are two issues I see:

- 1) The actual algorithm doesn't use a large number of points to estimate the NRCS. Only five points are used.
- 2) The standard deviation of the differences (over a large number of differences) is likely not a good estimate for the uncertainty in a single estimated NRCS.

I think both of these issues would contribute to substantially underestimating the uncertainty. I am skeptical that the uncertainties in estimated PIA could actually be as small as 0.25 to 0.5 dB when the calibration points are 100 to 200 km away.

An approach that would more faithfully estimate the error resulting from the single application of the algorithm on a single profile would be to:

- 1) Compute the difference using only five calibration points (as is done in the algorithm) selected randomly, and produce multiple realizations of each difference.
- 2) For each wind speed and distance bin, estimate the uncertainty by using the absolute value of the differences in that bin and taking the expected value or mean of those absolute differences.

Since the PIA uncertainty has direct influence on the uncertainties in hydrometeor retrievals, it is important that the NRCS uncertainty be estimated accurately. I'd like to see a more rigorous estimate of NRCS uncertainty used in the article.

Below are comments about details of the manuscript followed by editing corrections.

L38-39: Please clarify here. Are you referring specifically to liquid precipitation particles, or does this hold for both ice and liquid particles. If only liquid, it would be more clear to say "rain" rather than "precipitation particles".

L106: In what way has N=5 been tested and determined to be "optimal"? I will see if Section 2.3 sheds light on this.

L133-L136: There are a couple of statements here that say that interpolation yeilds sigma\_0 values with "lower" uncertainties. Lower than what? It is not clear if you are comparing against uncertainties in other parts of the distance-windspeed space, or if you are comparing against the model-driven method.

L136: This statement doesn't seem correct based on Figure 1. When wind speeds are small, the PIA uncertainties appear generally to be larger than about 3 dB, regardless of distance. For these windspeeds, it is only for the cases of very small distances are the PIA uncertainties very small.

L144: Using sigma to represent both surface backscatter cross section \*and\* uncertainties most likely will lead to some confusion. Consider using "s" to represent uncertainty - it is not uncommon to do this.

L161-L163: I think it is not accurate to say that equation 6 "accounts" for the modulation due to the surface properties. If anything, equation 6 ignores the dependency on surface properties, opting instead to simply weight the calibration point sigma\_0s based on distance from the profile of interest and to compute the uncertainty using the distance-based weights.

L171: There is no prior description or reference for X-MET.

L190: Is this correct? As written, if f\_surf is negative, r\_surf is larger (farther away) than r(n\_surf). This seems to conflict with what is stated in L187-L188.

L222-L226 and Figure 3, caption: Per Figure 3, there are broad areas of the midlatitude storm tracks, subtropics and tropics where calibration point fraction (CPF) is 0.7 or larger. Per the definition, the CPF is the "ratio of valid calibration points to the total number of radar profiles within each 1-deg x 1-deg grid cell". My understanding, then, is that 70% or more of the observed radar profiles in these regions are valid calibration points, which are clear profiles. That seems like a very large clear fraction. If my understanding isn't correct, some additional explanation is needed in the text.

L230: Bibliographic information is not provided for the works by Cox and Munk, Wu, and Freilich and Vanhoff. Please make sure your bibliography is complete.

L246-L49: This seems inconsistent with the results shown by Haynes et al. (2009), their Figure 3. At small wind speeds, they found the standard deviations of sigma\_0 to range up to 2.3-2.6 dB, depending on SST. The standard deviations presented in this work, as described for Figure 5, are substantially smaller. The standard deviations in Figure 5 seem inconsistent with Figure 4 of this work also, where the 25th and 75th percentile ranges are about +/- 5 dB at small wind speeds. Please provide some explanation and provide commentary in the text.

L258, Figure 6: As noted earlier, bib info has been omitted for Cox and Munk, Wu, and Freilich and Vanhoff.

L281-L282: I think "extensive" might be a better description than "persistent". While I agree that this stratocumulus deck likely \*is\* persistent (long in time duration), that can't be deduced from the radar observation.

Figure 10, caption: The second and third sentences are partial duplicates.

I won't comment on these further, but please make sure articles ("a", "an",

"the") are used where needed in the text. This seems to be an issue starting mainly in Section 4.

L319-L321: Averaging using further-removed calibration points may reduce the \*occurrence\* of transitions, but this is likely at the expense of accuracy. It would be appropriate (and fair, I think) to show sigma\_0,gas results from the EarthCARE approach in which there are transitions between the model-based and interpolation methodologies. I suspect there are similar non-physical jumps in those results.

L352-L356: I don't see the logical path by which the positive bias of EarthCARE's PIA estimate relative to CloudSat's would be due to differences in the frequencies with which each use the Wind/SST method. More explanation is needed here about how that conclusion was reached.

L359: I think you should apply the same adjective used when this result was presented at lines 334-335: "slightly"

L388-L390: See my prior comment regarding L352-L356 and adjust this text to match changes made there.

**Editing comments:**

L77: "selection" should be "selecting".

L112: "chose" should be "chosen".

L164: "it's" should be "its".

L227: "Section" should be "section".

L243, Figure 4: The color bar is unlabeled.

L250, Figure 5: Again, the color bar is unlabeled.

L298-L299: "at cloudy region" might be better as "at the cloudy profile".

L300: "and second one" should be "and the second one".

L301: "surrounding cloudy profile" should be "surrounding the cloudy profile".

L326: "Section" should be "section".

---

## Author Comment (AC1)

**Response to Referees and Reviewers**

September 30, 2025

**Manuscript egusphere-2025-3573 "The Estimation of Path Integrated Attenuation for the EarthCARE Cloud Profiling Radar"**

**1 Referee Comments (RC1)-Matthew Lebsock:**

1. Line 42: Add Lebsock and Suzuki 2016 DOI:10.1175/JTECH-D-16-0023.1.

   Thank you for pointing this out. Added the reference in the revised manuscript.

2. Equation 6: I agree that the fact that there are differences in the equation is an advantage IF the water vapor profile and the surface wind speed are approximately equal at points $x$ and $x_1$. However, in the presence of cloud at point x and clear sky 'calibration point' at $x_1$ we don't expect this to necessarily be true, especially for small scale unresolved by the model fields from which water vapor is derived. For example, Lebsock and Suzuki 2016 show using an LES that water vapor attenuation is larger in the cloudy targets than the clear targets, which makes physical sense.

   Thank you for your comment. We understand that Equation 6 relies on modeled water vapor and surface conditions, so its accuracy depends on how well the model represents these fields at both points. Small-scale differences, such as higher water vapor over clouds compared to clear-sky points, may lead to slight underestimation of hydro attenuation. This limitation is common to any PIA retrieval method, because the gas attenuation has to be computed anyhow (which needs an assumption on water vapour profiles).

3. Line 112: 'chose' → 'chosen'.

   Corrected in revised manuscript, Thank you.

4. Figure 1: I need help with this figure. First, I think you should show another panel with both the 'model-method' and 'interpolation-method' error plotted as a function of wind speed. Second, I think you should label on the existing panel which region is best for each method for clarity. Third, I can't quite understand why the interpolation method is better for a much greater distance between $x$ and $x_1$ when the wind speed at $x_1$ drops below about 3 m/s. I actually would expect the opposite – that the interpolation would work better over greater distances for higher wind speeds . Fourth, the residuals should be a function of both the wind speed at x and the wind speed at $x_1$ since they each influence one of the $\sigma_{0e}$ terms. Can you comment on points 3 and 4?

   Thank you for this suggestion.In response, we conducted additional analysis to better characterize PIA uncertainty.

   Regarding the influence of calibration-point wind speed, we agree that in principle the residuals depend on both the wind speed at the cloudy profile ($x$) and at the calibration point ($x_1$), since each influences one of the $\sigma_{0e}$ terms. o capture this, we segregated residuals by wind speed at $x$ and $x_1$ and constructed two separate PIA uncertainty lookup tables (LUTs):

   - Using calibration points for which $|\text{Wind}(x) - \text{Wind}(x_1)| \leq 2$ m/s (Fig. 1).
   - Using calibration points for which $2$ m/s $< |\text{Wind}(x) - \text{Wind}(x_1)| \leq 4$ m/s (Fig. 2).

   The regions in the LUT plots where each method performs best are labeled.

[Figure]

Figure 1: Lookup table of PIA uncertainty based on calibration points absolute wind speed difference within 2 m/s from the cloudy profile. The x-axis shows the wind speed of the cloudy profile, and the y-axis indicates the distance to the calibration point. The black line marks the boundary beyond which the model-driven method is preferred.

[Figure]

Figure 2: Lookup table of PIA uncertainty based on calibration points with an absolute wind speed difference greater than 2 m/s but no more than 4 m/s from the cloudy profile. The x-axis shows the wind speed of the cloudy profile, and the y-axis indicates the distance to the calibration point. The black line marks the boundary beyond which the model-driven method is preferred.

Within ∼85 km of the cloudy profile, winds are generally correlated, and most calibration points fall within ±2 m/s limits.

To address point 1, we are including two panels comparing PIA uncertainty from the model-driven and interpolation methods as a function of the cloudy profile wind speed:

- Using calibration points for which $|\text{Wind}(x) - \text{Wind}(x_1)| \leq 2$ m/s, for varying distances. (Fig. 3).
- Using calibration points for which $2$ m/s $< |\text{Wind}(x) - \text{Wind}(x_1)| \leq 4$ m/s, for varying distances. (Fig. 4).

[Figure]

Figure 3: PIA uncertainty from the model and clear-sky interpolation using calibration points with an absolute wind speed difference within 2 m/s from the cloudy profile, for varying calibration point distances. X-axis: wind speed of the cloudy profile; Y-axis: PIA uncertainty.

[Figure]

Figure 4: PIA uncertainty from the model and clear-sky interpolation using calibration points with an absolute wind speed difference greater than 2 m/s but no more than 4 m/s from cloudy profile, for varying calibration point distances. X-axis: wind speed of the cloudy profile; Y-axis: PIA uncertainty.

According to Fig.3, when calibration points within ±2 m/s are used, interpolation with clear-sky profiles generally yields lower uncertainty, particularly within 50 km, and the advantage is most pronounced at low wind speeds (<4 m/s). As distance increases, interpolation's advantage diminishes, and uncertainty converges to the model-driven method.

According to Fig.4, using calibration points with an absolute wind speed difference of 2-4 m/s from the cloudy profile, interpolation generally performs worse than the model, except at very low winds (<2.5 m/s), where it shows lower uncertainty than both the ±2 m/s case and the model. In all other cases, the model-driven method is preferred.

Comparison with the previous PIA uncertainty LUT, which included all calibration points, shows the following:

- Using only calibration points within ±2 m/s allows interpolation over larger distances at high wind speeds (>5 m/s) compared to the previous LUT (150–250 km vs 100–200 km), due to stronger wind correlations.

- At low wind speeds, uncertainty is higher, and the previously observed feature, interpolation over thousands of kilometers, is no longer present. This arises from calibration-point selection: using points with absolute wind speed difference more than $\pm 2$ m/s relative to the cloudy profile generally produces lower uncertainty at larger distances in low-wind regimes. The previous LUT included all calibration points, effectively incorporating this feature and allowing interpolation over greater distances. Generally, as distance to calibration point increases, at low wind speed regimes, the uncertainties of both the interpolation and model-driven approaches are similar, typically in the 3-4 dB which reflects the fact that both approaches perform similarly poorly under such conditions.

Finally, although residuals depend on both the wind speed of the cloudy profile and the calibration point, our analysis shows that PIA uncertainty is mainly controlled by the distance to the calibration point. At short distances, winds are more similar, reducing uncertainty, while at larger distances, divergence increases. Most data within the allowed interpolation range fall within $\pm 2$ m/s of the cloudy profile wind speed, so distance remains the dominant factor, as reflected in the previous PIA uncertainty look-up table.

5. Line 159: Related to point above about low wind speeds here you say you exclude the low wind speeds from interpolation which is what I would expect. 'In contrast, the method used here allows interpolation even when the calibration points are 200 km to 100 km from the cloudy pixel in wind speed conditions between 4 and 15 m/s'.

In practice, interpolation is not applied at very large calibration distances in low wind speed regimes, since in these conditions the uncertainty from both methods is already similar. Extending interpolation to such large distances can actually degrade the performance.

6. Line 164: I think you will get an even better uncertainty estimate if you bin by wind speed at both x and x1. 'Each calibration point used in the PIA estimation is weighted based not only on it's distance from the point of interest 165 but also on the potential uncertainty associated with wind speed at that location'.

We agree that both wind speeds (at $x$ and $x_1$) can influence the uncertainty. However, our analysis showed very similar behavior when segregating by both variables, with distance to the calibration point remaining the dominant factor, as already summarized in the figure.

7. Equation 14: Several terms are not defined: lambda, c, tau_p.

Thank you for the comment. We have defined all previously undefined terms in Equation 14.

8. Lines 288-300: The 'model' used in precip-column is actually an empirical look-up-table derived from clear sky observations not the li model. 'The first approach, referred to as the Wind/SST method, estimates the NRCS at cloudy region in absence of hydrometeor and presence of gaseous attenuation ($\sigma$ gas 0 ) as a function of surface wind speed and SST using geophysical models (Li et al., 2005)'.

We acknowledge this correction. The initial description followed the 2C-PRECIP-COLUMN documentation, but in the revised manuscript we will clarify that the approach relies on an empirical look-up table derived from clear-sky observations, and not directly on the Li et al. (2005) geophysical model.

---

## Author Comment (AC2)

**Response to Referees and Reviewers**

November 13, 2025

Manuscript egusphere-2025-3573 "The Estimation of Path Integrated Attenuation for the EarthCARE Cloud Profiling Radar"

**1 Referee Comments (RC2):**

The authors describe a method for estimating the path integrated attenuation due to hydrometeors in an observed profile. The method uses two techniques, the first which uses measurements of surface backscatter in clear-sky profiles in the vicinity of the observed profile, and the second which used a scattering model to estimate the surface backscatter at the location of the observed profile. While CloudSat's 2C-RAIN-PROFILE product adopted a similar method, in this method the clear-sky profiles are allowed to be much further from the observed profile (~30 km for CloudSat versus 100-200 km for this method).

The paper is useful to document the methods used for EarthCARE products and so is likely appropriate for an EarthCARE special issue. I have one primary concern and that is that I think the uncertainty for the retrieved PIA has been estimated improperly. This estimation is described in lines 109 to 143.

In their approach, the authors compile a "large set" of clear-sky profiles. For a selected profile, differences described in equation (8) are computed against every other profile in the dataset and then decomposed by wind speed and separation distance. For each wind speed and distance bin, the standard deviation of the differences is taken, computing over all of the data points in the bin.

**1. There are two issues I see:**

- 1) The actual algorithm doesn't use a large number of points to estimate the NRCS. Only five points are used.
- 2) The standard deviation of the differences (over a large number of differences) is likely not a good estimate for the uncertainty in a single estimated NRCS.
- I think both of these issues would contribute to substantially underestimating the uncertainty. I am skeptical that the uncertainties in estimated PIA could actually be as small as 0.25 to 0.5 dB when the calibration points are 100 to 200 km away.
- An approach that would more faithfully estimate the error resulting from the single application of the algorithm on a single profile would be to:
- 1) Compute the difference using only five calibration points (as is done in the algorithm) selected randomly, and produce multiple realizations of each difference.
- 2) For each wind speed and distance bin, estimate the uncertainty by using the absolute value of the differences in that bin and taking the expected value or mean of those absolute differences. Since the PIA uncertainty has direct influence on the uncertainties in hydrometeor retrievals, it is important that the NRCS uncertainty be estimated accurately. I'd like to see a more rigorous estimate of NRCS uncertainty used in the article.

Thank you for the comments. I would like to clarify the method used to estimate the PIA uncertainty and how it is applied into our analysis.

In the implementation of the methodology, the five calibration points can be located at varying distances from the cloudy profile. The contribution of each calibration point to the total error is weighted based on its distance from the cloudy profile (with nearer points receiving higher weights) using the PIA uncertainty look-up table (LUT), see Figure 1 in the manuscript.

The uncertainty LUT provides the uncertainty associated to a given wind speed (or equivalently a given  $\sigma_0$ ) in correspondence to a fixed separation distance of the calibration point.

To quantify this uncertainty, a large set of clear-sky profiles with measured NRCS ( $\sigma_0^{gas}$ ) and wind speed is compiled based on one year of EarthCARE observations.

All the profiles with wind speed  $u_j$  in a given interval j  $((j-1)\delta u

Figure 1: PIA uncertainty Look-up-table created using mean of absolute differences of residuals. The solid black contour delineates the transition boundary beyond which the model-driven method yields lower PIA uncertainty.

Regarding the concern about unrealistically low uncertainties when calibration points are located 100–200 km away, I would like to clarify that the uncertainty is highly dependent on wind speed. For calibration points within 100 km, the uncertainty is approximately 0.8

Figure 2: PIA uncertainty as a function of wind speed at the cloudy profile for different separation distances between calibration and cloudy points.

dB at higher wind speeds (8-15 m/s) and increases to about 1-3.5 dB at lower wind speeds (4-0.5 m/s). Refer Fig.(2) for clarification.

Below are comments about details of the manuscript followed by editing corrections.

2. L38-39: Please clarify here. Are you referring specifically to liquid precipitation particles, or does this hold for both ice and liquid particles. If only liquid, it would be more clear to say "rain" rather than "precipitation particles".

Thank you for the comment. We are referring specifically to liquid precipitation. We will revise the text in the new version to use "rain" instead of "precipitation particles" for clarity.

3. L106: In what way has N=5 been tested and determined to be "optimal"? I will see if Section 2.3 sheds light on this.

We selected N=5 based on practical considerations. Calibration points are determined following the criteria described in Section 2.3: at least six clear profiles must be present within a 10 km along-track segment, the standard deviation of the measured NRCS must be less than 0.3 dB, and the calibration NRCS is defined as the mean NRCS of those clear profiles. When selecting five calibration points, we maintain a 10 km separation between each, ensuring that no two points fall within the same 10 km segment used for NRCS averaging. Consequently, when the cloudy region is adjacent to a continuous clear-sky area, the farthest calibration point is located approximately 50 km away. We also tested configurations using 5, 10, and 15 calibration points, and the resulting differences in PIA estimates were marginal (Mean of 0 dB and standard deviation of 0.05 dB), using five calibration points provided a good balance between accuracy and computational efficiency, and was therefore selected as optimal. The corresponding Fig.(3) is attached for reference.

Figure 3: Differences in PIA estimates obtained using 5 calibration points, compared to those derived using 10 and 15 calibration points.

4. L133-L136: There are a couple of statements here that say that interpolation yields sigma\_0 values with "lower" uncertainties. Lower than what? It is not clear if you are comparing against uncertainties in other parts of the distance-windspeed space, or if you are comparing against the model-driven method.

Thank you for your comment. The comparison refers to the uncertainties from the model-driven method. We will clarify this in the revised manuscript.

5. L136: This statement doesn't seem correct based on Figure 1. When wind speeds are small, the PIA uncertainties appear generally to be larger than about 3 dB, regardless of distance. For these windspeeds, it is only for the cases of very small distances are the PIA uncertainties very small.

Thank you for your comment. At low wind speeds, PIA uncertainties are generally large for both clear-sky interpolation and the model. The statement that interpolation yields lower uncertainty refers specifically to a comparison with the model, though the difference is marginal. This will be clarified in the revised manuscript.

6. L144: Using sigma to represent both surface backscatter cross section \*and\* uncertainties most likely will lead to some confusion. Consider using "s" to represent uncertainty - it is not uncommon to do this.

Thank you for the comment. We will use a different symbol to represent the uncertainty.

7. L161-L163: I think it is not accurate to say that equation 6 "accounts" for the modulation due to the surface properties. If anything, equation 6 ignores the dependency on surface properties, opting instead to simply weight the calibration point sigma\_0s based on distance from the profile of interest and to compute the uncertainty using the distance-based weights.

Thank you for your comment. The statement refers to the fact that Eq.(6) (refer the manuscript) includes  $\sigma_{0e}$ , the effective NRCS, which represents the surface backscatter in the absence of atmospheric and gas attenuation and is inherently a function of surface wind speed and SST. Therefore, the influence of surface properties is implicitly accounted for through  $\sigma_{0e}$  in the formulation.

8. L171: There is no prior description or reference for X-MET.

Thank you for the comment. We will provide a clearer description in the revised manuscript.

9. L190: Is this correct? As written, if f\_surf is negative, r\_surf is larger (farther away) than r(n\_surf). This seems to conflict with what is stated in L187-L188.

Thank you for pointing this. The description in L187–L188 was inaccurate, and we will revise it for consistency and clarity in the revised manuscript.

10. L222-L226 and Figure 3, caption: Per Figure 3, there are broad areas of the midlatitude storm tracks, subtropics and tropics where calibration point fraction (CPF) is 0.7 or larger. Per the definition, the CPF is the "ratio of valid calibration points to the total number of radar profiles within each 1-deg x 1-deg grid cell". My understanding, then, is that 70 or more of the observed radar profiles in these regions are valid calibration points, which are clear profiles. That seems like a very large clear fraction. If my understanding isn't correct, some additional explanation is needed in the text.

Thanks for the comment. To clarify, the calibration point fraction (CPF) shown in Figure 3 (refer the manuscript) represents the six-month mean fraction of profiles within each 1-deg x 1-deg grid cell cell that satisfy the calibration point criteria described in Section 2.3 (which will be better explained in the revised manuscript). Therefore it is an average number. If we compute the "clear-sky plus ice-cloud only" fraction from the EarthCARE CPR data it can indeed reach values as high as 70% in the tropical regions in the sinking branches of the Hadley cell (Fig.4). The clear-sky and ice-cloud only profiles are identifies using significant detection mask in EarthCARE and to identify ice-clouds, a cloud base temperature of less than -263.15K is imposed. Of course this value would be reduced when considering other observations that include visible and infrared radiation (thus more sensible to thinner clouds) like in (Leah Bertrand and de Boer, 2023).

11. L230: Bibliographic information is not provided for the works by Cox and Munk, Wu, and Freilich and Vanhoff. Please make sure your bibliography is complete.

Thank you for the comment. We will add complete bibliographic information for Cox and Munk, Wu, and Freilich and Vanhoff in the reference list.

Figure 4: Global distribution of the clear profiles and ice-cloud-only fraction within each  $1^{\circ} \times 1^{\circ}$  grid cell.

12. L246-L49: This seems inconsistent with the results shown by Haynes et al.(2009), their Figure 3. At small wind speeds, they found the standard deviations of sigma\_0 to range up to 2.3-2.6 dB, depending on SST. The standard deviations presented in this work, as described for Figure 5, are substantially smaller. The standard deviations in Figure 5 seem inconsistent with Figure 4 of this work also, where the 25th and 75th percentile ranges are about +/- 5 dB at small wind speeds. Please provide some explanation and provide commentary in the text.

Thank you for the comment. The apparent discrepancy arises because Figure 4 and Figure 5 in the manuscript represent different types of variability. In Figure 4 (refer the manuscript) the error bars show the standard deviation of NRCS within each wind-speed bin (aggregated across all SST values). These wind-bin standard deviations range from 0.4–0.5 dB at high wind speeds (8-12 m/s) to 1–4 dB at low wind speeds (4-0.5 m/s). These values are generally consistent with the results in Haynes et al.(2009). By contrast, Figure 5 (refer the manuscript) displays the distribution of standard deviations calculated for individual 10 km along-track clear-sky segments, and we plot the median of those per-segment standard deviations within each wind bin. This standard deviation is much smaller because it reflects variability within short homogeneous clear-sky segments, not across the full population of scenes.

13. L258, Figure 6: As noted earlier, bib info has been omitted for Cox and Munk, Wu, and Freilich and Vanhoff.

Thank you for your comment.

14. L281-L282: I think "extensive" might be a better description than "persistent". While I agree that this stratocumulus deck likely \*is\* persistent (long in time duration), that can't be deduced from the radar observation.

Thank you for the comment. We agree and will use "extensive" instead in the revised manuscript.

15. Figure 10, caption: The second and third sentences are partial duplicates. I won't comment on these further, but please make sure articles ("a", "an", "the") are used where needed in the text. This seems to be an issue starting mainly in Section 4.

Thank you for the comment. We will remove the duplicate sentences in the caption and review the text to ensure correct use of articles throughout the manuscript.

16. L319-L321: Averaging using further-removed calibration points may reduce the \*occurrence\* of transitions, but this is likely at the expense of accuracy. It would be appropriate (and fair, I think) to show sigma\_0,gas results from the EarthCARE approach in which there are transitions between the model-based and interpolation methodologies. I suspect there are similar non-physical jumps in those results.

Thank you for the comment. As noted in the manuscript, the EarthCARE approach also exhibits transitions when switching between the methods. We will include an example case demonstrating the corresponding jumps in the EarthCARE-based estimate for comparison in the revised manuscript.

17. L352-L356: I don't see the logical path by which the positive bias of EarthCARE's PIA estimate relative to CloudSat's would be due to differences in the frequencies with which each use the Wind/SST method. More explanation is needed here about how that conclusion was reached.

Thank you for your comment. We agree that the explanation for the positive bias requires further clarification. We are currently investigating this aspect in more detail and are in communication with the CloudSat team to ensure that the correct datasets and methodologies are being used for the comparison. We will provide a more detailed explanation and updated discussion in the revised manuscript.

18. L359: I think you should apply the same adjective used when this result was presented at lines 334-335: "slightly"

Thank you for the comment. We will change the adjective accordingly to maintain consistency in the manuscript.

19. L388-L390: See my prior comment regarding L352-L356 and adjust this text to match changes made there.

Thank you for your comment.

Editing comments:

- 20. L77: "selection" should be "selecting".
- 21. L112: "chose" should be "chosen".
- 22. L164: "it's" should be "its".
- 23. L227: "Section" should be "section".
- 24. L243, Figure 4: The color bar is unlabeled.
- 25. L250, Figure 5: Again, the color bar is unlabeled.
- 26. L298-L299: "at cloudy region" might be better as "at the cloudy profile".
- 27. L300: "and second one" should be "and the second one".
- 28. L301: "surrounding cloudy profile" should be "surrounding the cloudy profile".
- 29. L326: "Section" should be "section".

Thank you for all the editing comments. We will correct these issues in the revised manuscript.

**References**

John Haynes Leah Bertrand, Jennifer E. Kay and Gijs de Boer. A global gridded dataset for cloud vertical structure from combined cloudsat and calipso observations. *Earth System Science Data*, 2023. doi: 10.5194/essd-16-1301-2024. URL https://essd.copernicus.org/articles/16/1301/2024/.